

# Analysis of autogenic bifurcation processes resulting in river avulsion

Gabriele Barile[1], Marco Redolfi[1], and Marco Tubino[1]

[1]Department of Civil, Environmental and Mechanical Engineering, via Mesiano 77, 38123 Trento (TN), Italy

**Correspondence:** Gabriele Barile (gabriele.barile@unitn.it)

**Abstract.** River bifurcations are constituent components of multi-thread fluvial systems, playing a crucial role in their morphodynamic evolution and the partitioning of water and sediment. Although many studies have been directed at exploring bifurcation dynamics, the conditions under which avulsions occur, resulting in the complete abandonment of one branch, are still not well understood. To address this knowledge gap, we develop a novel 1D numerical model, based on existing nodal

point relations for sediment partitioning, which allows for the simulation of the morphodynamic evolution of a free bifurcation. Model results show that when the discharge asymmetry is so high that the shoaling branch does not transport sediments (partial avulsion conditions) the dominant branch undergoes significant degradation, leading to a higher inlet step between the bifurcates and further amplifying the discharge asymmetry. The degree of asymmetry is found to increase with the length of the downstream channels, to the point that when they are sufficiently long, the shoaling branch is completely abandoned (full

avulsion conditions). To complement our numerical findings, we also formulate a new analytical model that is able to reproduce the essential characteristics of the partial avulsion equilibrium, which enables us to identify the key parameters that control the transition between different configurations. In summary, this research sheds light on the fundamental processes that drive avulsion through the abandonment of river bifurcations. The insights gained from this study provide a foundation for further investigations and may offer valuable information for the design of sustainable river restoration projects.

## 1   Introduction

Bifurcations constitute essential geomorphic elements of a variety of fluvial systems, such as braiding rivers and deltas. The morphodynamic evolution of these systems strongly depends on the water and sediment partitioning set by the bifurcation node, which in turn is determined by the reach-scale morphodynamic processes occurring in the three branches connected by the bifurcation node. Understanding these processes is therefore key to foresee the behaviour of multi-thread systems across

temporal scales, as well as to design sustainable river restoration projects (Habersack and Piégay, 2007).

Among the morphodynamic processes that occur at bifurcations, the partial or complete abandonment of one of the bifurcates is of crucial importance. This event, which leads to the diversion of the majority or all of the incoming water and sediment discharges towards one of the two bifurcates, can be considered as one of the possible styles of river avulsions, such as "soft" or "choking" avulsions (Edmonds et al., 2011; Leddy et al., 1993). These processes have been shown to play a major role in

the evolutionary trajectories of braiding networks (Ferguson, 1993), lowland rivers (Slingerland and Smith, 2004), river deltas





(Salter et al., 2018) and alluvial fans (Field, 2001), as they can cause shifts in evolving morphologies and suddenly modify the flooding risk and water resource availability in the downstream areas. Moreover, a clear understanding of the conditions that can ensure the long-term preservation of bifurcating systems is also crucial for the effective implementation of river interventions aimed at recovering the eco-morphological quality of river ecosystems, thus reducing the need for intensive maintenance.

Although many studies have been directed at exploring the possible long-term states of bifurcations, little is yet known about the conditions for which avulsions occur during the transient stage of an evolving bifurcation. Among others, 1-D approaches proved to be very effective in analysing the key mechanisms that govern water and sediment partitioning at bifurcation nodes. These studies stemmed largely from the analytical model developed by Bolla Pittaluga et al. (2003), hereafter referred to as BRT, where a quasi-2D physically-based nodal relationship was used to describe the sediment partitioning at the node.

The BRT model was originally formulated to analyse the dependence on the parameters of the upstream flow of the long-term equilibrium states of a "free" bifurcation (i.e. in the absence of external factors influencing the morphodynamic processes occurring at the node). The model was further extended to include the effects of migrating bars (Bertoldi et al., 2009), tides (Ragno et al., 2020), curvature in the upstream channel (Kleinhans et al., 2012), slope advantage between the branches (Redolfi et al., 2019), and downstream effects generated by prograding branches (Salter et al., 2018).

The main outcome of the BRT model, later confirmed by the experimental results of Bertoldi and Tubino (2007), is the occurrence of two distinct regimes, depending on the values of the half-width to depth ratio $\beta$ (hereafter referred to as "aspect ratio") and the Shields stress $\theta$ of the upstream flow. Specifically, for low values of the aspect ratio $\beta$ the model only admits of one stable equilibrium solution, in which water and sediments are partitioned evenly by the bifurcation node; in other words, the only equilibrium state of the bifurcation is "balanced". When the aspect ratio $\beta$ exceeds a critical threshold, the balanced

solution becomes unstable and the system admits of two stable, anti-symmetric and unbalanced equilibrium configurations. In these unbalanced equilibrium states, both branches are morphodynamically active, i.e. they have a non-negligible transport capacity. As highlighted by Redolfi et al. (2016), the critical aspect ratio is found to coincide with the resonant value that discriminates between prevailing upstream or downstream propagation of morphological changes (Zolezzi and Seminara, 2001), which reveals the close connection between the dynamics of free bifurcations and the framework of 2-D morphodynamic

influence.

However, when further increasing the value of $\beta$ a second threshold appears, beyond which all stable equilibrium states identified by the BRT model are characterised by a branch with vanishing transport capacity (see Fig. 1). These states correspond to bifurcations in which one of the two branches is morphodynamically inactive, thus more likely to be abandoned as the bifurcation evolves. Under these conditions the theoretical predictions based on the BRT model are no longer suitable, as

the model assumes that both branches are able to adjust their bed slope to a prescribed equilibrium value, which is clearly not possible for the channel that is no longer transporting sediments. The question then arises on whether in this condition the bifurcation can still attain an equilibrium state, and to what extent such state differs from that predicted by the BRT model. To address this question, an evolutionary model is required to trace the trajectory of a bifurcation starting from a nearly-balanced initial state.





In the past two decades, many authors attempted at investigating the transient behaviour of bifurcations belonging to different fluvial environments, though most of these studies were focused on the effect of external forcing factors. For example, Salter et al. (2018) developed a numerical model to study the transient behaviour of bifurcations with prograding branches, considering suitable initial and boundary conditions for reproducing the evolutionary trajectories of bifurcations in depositional environments. Using a similar approach, Iwantoro et al. (2022) studied the effect of tides on the stability of bifurcations
by properly varying the downstream boundary conditions. Other evolutionary models of bifurcations available in the literature were tailored to specific case studies, such as bifurcations located downstream a reservoir (Mendoza et al., 2022) and longitudinal training walls in presence of alternate bars in the upstream channel (Le et al., 2018).

The stability of equilibrium states of free bifurcations was indeed studied by Edmonds and Slingerland (2008). However, their analysis was limited to specific deltaic environments in which both branches of the bifurcation were always morpho-
dynamically active. On the contrary, in some of the 3-D numerical simulations performed by Kleinhans et al. (2008) all the incoming solid discharge was steered towards one of the two downstream branches. Analysing these simulations, the authors highlighted the limited capability of the BRT model – both in the original version and with the modifications proposed in their study – to correctly predict the equilibrium value of the water discharge asymmetry of the bifurcation. They specifically showed that bifurcations can display a much higher degree of asymmetry than that foreseen by BRT. Although the simulations consid-
ered a bifurcation with a curved upstream channel, thus introducing an external forcing in the model, this study highlighted the important role that the transient behaviour may have in determining the long-term equilibrium state of a bifurcation.

There is therefore a lack of knowledge about the transient behaviour of a "free" bifurcation and about the possible existence of an autogenic mechanism that may lead to the partial or complete abandonment of a branch. In this work, we aim at analysing the transient behaviour of an initially balanced bifurcation that spontaneously evolves over time. To this purpose, we employ
a novel 1D numerical model that relies on the BRT relationship for sediment partitioning at the bifurcation node and accounts for the morphodynamic evolution of the three branches composing the bifurcation. By analysing the model outputs, we firstly outline the evolutionary processes that lead the system to reach its long-term, unbalanced equilibrium configuration. Secondly, we show how model results vary considerably according to the aforementioned regions of the BRT model parameter space, displayed in Fig. 1. Specifically, we show how the bifurcation may evolve towards considerably more unbalanced equilib-
rium states than those foreseen by the BRT model, including those in which one branch is completely abandoned. Lastly, we formulate a simple analytical model to characterise this new category of equilibrium states.

## 2   Methods

### 2.1   Formulation

We consider a "free" bifurcation consisting of three branches, each characterised by a rectangular cross-section and fixed banks.
The widths of the bifurcates $b$ and $c$ are assumed to be both equal to half the width of the upstream branch $a$. The upstream channel is fed by constant liquid ($Q_0$) and solid ($Q_{s0}$) discharges, while at the outlet of the downstream branches $b$ and $c$ a constant water-surface elevation $h_d$ is set. At the bifurcation node we also impose the water discharge balance and an equal





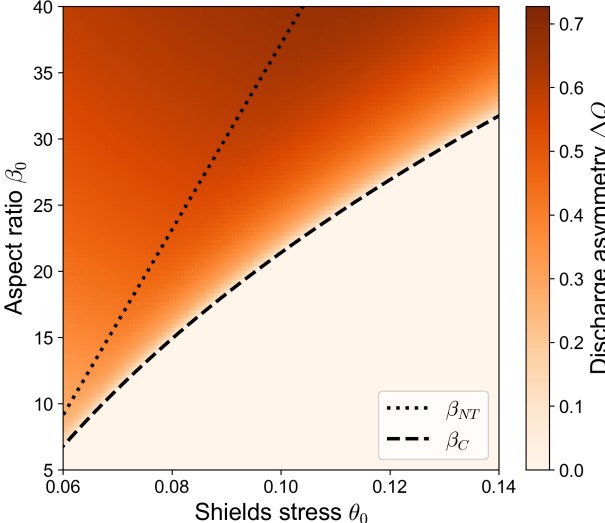

**Figure 1.** Regions of the parameter space identifying different types of equilibrium solutions of river bifurcations computed by means of the BRT model (Bolla Pittaluga et al., 2003) for different combinations of the aspect ratio $\beta_0$ and Shields stress $\theta_0$ of the upstream channel. The dashed line indicates the critical aspect ratio $\beta_C$, which separates the region where only balanced equilibrium configurations are stable (lower right region of the plot) from the region where only unbalanced equilibrium configurations are stable (upper left portion of the plot). The dotted line indicates the no-transport aspect ratio $\beta_{NT}$: when $\beta_0 \geq \beta_{NT}$, the BRT model predicts a vanishing transport capacity in the shoaling branch. Darker shades indicate larger values of the equilibrium discharge asymmetry $\Delta Q$, defined as the difference in the water discharges flowing in the bifurcates scaled by the incoming discharge.

water level for the three branches. Lastly, we employ the quasi-2D two-cell nodal condition proposed by Bolla Pittaluga et al. (2003) to compute the transverse exchange of sediments at the node.

To model the morphodynamic evolution of the bifurcation over time, we solve numerically the gradually varied flow equation and the Exner equation along the three branches, according to a quasi-steady approach. As initial conditions we consider a balanced bifurcation, where water and sediments are partitioned equally among the downstream branches. After applying a small perturbation on the transverse bed slope between the branches, we then let the bifurcation evolve spontaneously over time.

**Computation of hydrodynamic conditions and morphological evolution along the branches**

The hydrodynamic conditions are evaluated by assuming that riverbed changes are much slower than the flow adaptation, which implies that the solution for the liquid phase can be decoupled from that of the solid phase. The constancy of the water supply then implies that a unique value of water discharge can be considered for each branch. The water-surface profile computation



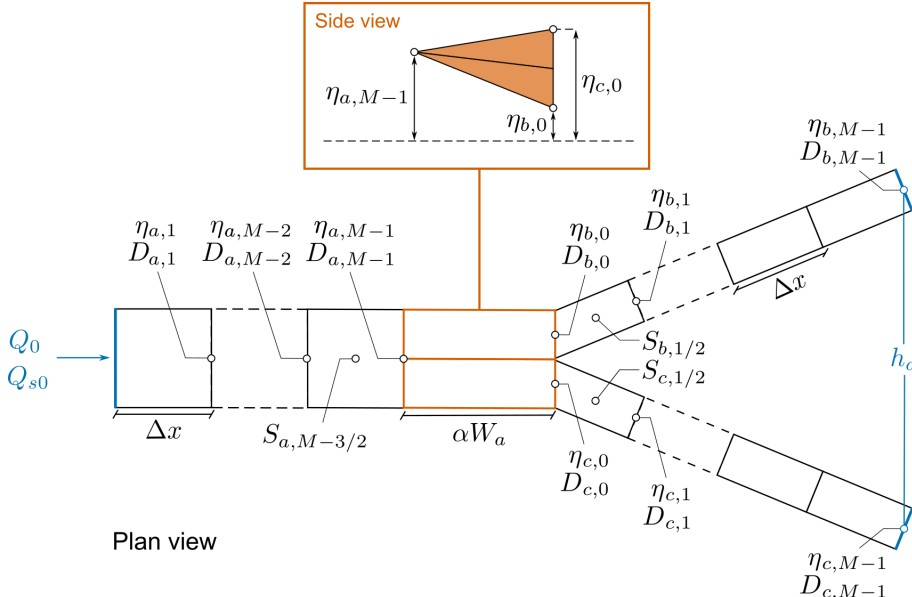

**Figure 2.** Spatial discretization of the water depth $D$, the bed elevation $\eta$ and the bed slope $S$ along the channels and within the nodal bifurcation cells. The side view illustrates in detail the triangular step scheme adopted for the bed elevation profiles of the node cells, whose length is given by the product of the order-1 parameter $\alpha$ and the upstream channel width $W_a$. $Q_0$ and $Q_{s0}$ represent the water and sediment supply, while $h_d$ stands for the fixed water level at the downstream end of branches $b$ and $c$.

along each branch is then performed by integrating in the the $x$-direction the gradually varied flow equation:

$$\frac{\mathrm{d}D}{\mathrm{d}x} = \frac{S - j(Q,D)}{1 - Fr^2(Q,D)}, \tag{1}$$

where $D$ is the water depth, $S$ is the local bed slope, $j$ is the energy gradient, and $Fr$ is the Froude number. Both $j$ and $Fr$ are computed as functions of the water discharge $Q$ and of the water depth $D$ as:

$$j = \left(\frac{Q}{WC\sqrt{g}D^{3/2}}\right)^2, \tag{2a}$$

$$Fr = \frac{Q}{W\sqrt{g}D^{3/2}}, \tag{2b}$$

where $W$ is the width of the branch, $g$ is the gravity acceleration and $C$ is the dimensionless Chézy coefficient, which is here for simplicity considered to be independent of the water depth and known a priori.

Integrating Eq. (1) along the downstream bifurcates requires the knowledge of the respective flow rates, which we compute by means of an iterative procedure that relies on the assumption of equal surface elevation at the bifurcation node. This procedure is similar to that employed by previous models (e.g., Kleinhans et al., 2008), and will be later described in this section.

Once the water discharge and the water depth along the three branches have been determined, the evolution over time of the bed profile is computed by integrating in time the Exner equation, which for channels having a rectangular cross-section and





constant width is written in the form:

$$(1-p)\frac{\partial \eta}{\partial t} = -\frac{1}{W}\frac{\partial Q_s}{\partial x}, \tag{3}$$

where $t$ is the time, $\eta$ the local bed elevation, $p$ the bed porosity, and $Q_s$ the sediment flux. The latter is expressed as:

$$Q_s = W\sqrt{g\Delta D_s^3}\,\Phi(\theta), \tag{4}$$

where $\Delta$ is the relative submerged weight of sediments, $D_s$ is the median grain size, and $\Phi(\theta)$ is the dimensionless transport capacity per unit width. The latter is evaluated by means of a suitable transport formula (e.g., Meyer-Peter and Müller, 1948; Parker, 1978) as a function of the Shields stress $\theta$, namely:

$$\theta = \frac{jD}{\Delta D_s}. \tag{5}$$

**Water surface gradient and transverse solid discharge at the node cells**

Following the approach proposed by Bolla Pittaluga et al. (2003), we model the last reach of the upstream channel $a$ by means of two node cells, whose length is $\alpha W_a$ (where $\alpha$ is a order-1 parameter), which are allowed to exchange water and sediments in the transverse direction. For these node cells, the mathematical formulation must be amended to suitably represent the quasi-2D scheme introduced by BRT to account for the sediment transport redistribution driven by flow exchange and transverse bed slope.

The first question that arises is how to schematize the elevation gap between the two cells. Here we integrate the gradually varied flow equation and the Exner equation by considering the "triangular step" scheme represented in Fig. 2, i.e. by assuming that the bed elevation varies linearly along each node cell.

A second issue concerns how the water-surface profile is computed, and specifically how to determine the water level at the inlet of the nodal cells, which sets the boundary condition for the computation of the backwater profile along the upstream channel $a$. Most existing 1D models (e.g. Kleinhans et al., 2008; Iwantoro et al., 2022) solved this problem by assuming a constant water surface elevation throughout the bifurcation cells. However, considering that the nodal cells extend over several channel width, this may be not fully appropriate, especially in relatively steep channels. To overcome this limitation, the difference in water level between the upstream and downstream ends of the node cells is calculated by integrating Eq. (1) along the node cells themselves, assuming a composite cross-section made of two halves with different bed elevation. The resulting formula for the energy gradient $j$ reads:

$$j = \left(\frac{Q_a}{C\sqrt{g}W_a\dfrac{D_{ab}^{3/2}+D_{ac}^{3/2}}{2}}\right)^2, \tag{6}$$

where $Q_a\,(=Q_0)$ is the water discharge, $D_{ab}$ and $D_{ac}$ stand for the water depths of the two halves of the cross-section, and the Froude number is computed using the average water depth, i.e., $D = (D_{ab}+D_{ac})/2$.




A third problem to be addressed regards the computation of the sediment fluxes at the cell boundaries and the subsequent update of the bed elevation of the node cells. The latter is achieved by means of the Exner equation, which we use in integrated form for each node cell. To compute the transverse exchange of sediments between the node cells we employ the nodal relationship proposed by BRT, which reads:

$$Q_{sy} = Q_{sa}\left(\frac{Q_b - Q_c}{2Q_a} - \frac{2\alpha r}{\sqrt{\theta_a}}\frac{\eta_{bN} - \eta_{cN}}{W_a}\right),\tag{7}$$

where $r$ is an empirical coefficient that quantifies the effect of transverse bed slope (e.g. Baar et al., 2018), $Q_{sa}$ and $\theta_a$ are the sediment flux and the Shields stress of the incoming flow, $\eta_{bN}$ and $\eta_{cN}$ are the average bed elevations of the two node cells, and $Q_b$ and $Q_c$ are the water discharges flowing in the downstream branches. The transverse flux of sediments $Q_{sy}$ is then used to compute the evolution of the node cell elevation over time. To do so, we integrate the Exner equation (3) in space over the node cells. By considering the aforementioned "triangular step" geometry, the resulting equation for the node cell $b$ reads:

$$(1 - p)\frac{\mathrm{d}\eta_{bN}}{\mathrm{d}t}\frac{\alpha W_a^2}{2} + Q_{sb} - \frac{Q_{sa}}{2} - Q_{sy} = 0,\tag{8}$$

where $Q_{sb}$ is the sediment flux entering the channel $b$. An analogous equation is obtained for the node cell $c$.

## 2.2 Numerical scheme

In order to integrate the gradually-varied flow equation (1) and the Exner equation (3) over our domain, we discretize each of the three branches composing the bifurcation by means of $M$ nodes. As shown in Fig. 2, the bed elevation $\eta$, the water depth $D$, the Shields parameter $\theta$ and the solid discharge $Q_s$ are all defined at each node, identified by means of the index $i$. On the contrary, the longitudinal bed slope $S$ is defined between each couple of subsequent nodes.

In all simulations performed, the flow conditions along the three branches is subcritical at any point in time and space. Consequently, we integrate the gradually varied flow equation in upstream direction and use a backward difference scheme for the spatial derivative in the Exner equation, according to an upwind approach. In the following, some details about the integration procedures adopted in the numerical scheme are provided.

To solve numerically Eq. (1) between two adjacent nodes, we first discretize the derivative of water depth $D$ and bed elevation $\eta$ as:

$$\begin{aligned}\frac{\mathrm{d}D}{\mathrm{d}x} &= \frac{D_{k,i+1}^n - D_{k,i}^n}{\Delta x}, \\ S = S_{k,i+1/2}^n &= \frac{\eta_{k,i}^n - \eta_{k,i+1}^n}{\Delta x},\end{aligned} \qquad \begin{cases}k = a, b, c \\ i = 0..M - 1\end{cases}\tag{9}$$

where $\eta_{k,i}^n$ stands for the bed elevation at node $i$ at the generic time step $n$, and $\Delta x$ is the prescribed grid size.

We then solve Eq. (1) in upstream direction by means of a fourth-order Runge-Kutta scheme, computing the water depth at the node $i$ given the water depth at the subsequent node $i + 1$. The integration procedure starts from the water depth at the downstream end of branches $b$ and $c$, which is computed as the difference between the prescribed water level $h_d$ and the bed elevation at the last node of each branch resulting from the previous iteration. It is worth noting that, according to the adopted





quasi-steady approach, we use the bed slope $S$ computed at the previous time step $n$ to compute the water depth along each branch at the new time step $n + 1$.

The integration of the gradually varied flow equation eventually returns the water depths at the first nodes of branches $b$ and $c$ (i.e., $D_{b,0}$ and $D_{c,0}$). If the condition of equal water level at the bifurcation node isn't met, Eq. (1) is iteratively integrated along the branches $b$ and $c$ until the condition is satisfied. Iterations are performed over the water discharge flowing in one branch

according to the Newton-Raphson method. At each iteration, the discharge flowing in the other branch is retrieved from water mass continuity. The water discharges flowing in branches $b$ and $c$ resulting from these iterations are then used to compute the final water-surface profiles along the bifurcates.

To update the bed profile along the three branches, we integrate the Exner equation (3) in time, using a time step $\Delta t$ computed according to a CFL condition and an upwind scheme to discretize the spatial derivative of the sediment flux $Q_s$:

$$\eta_{k,i}^{n+1} = \eta_{k,i}^n - \frac{1}{(1-p)W_k}\frac{\Delta t}{\Delta x}\left(Q_{sk,i}^{n+1} - Q_{sk,i-1}^{n+1}\right) \quad \begin{cases} k = a, b, c \\ \\ i = 1..M-1 \end{cases}. \tag{10}$$

Note that Eq. (10) is used to update the bed elevation of all nodes except the first node of each channel. The first node of channel $a$ is updated by using a ghost cell, while the bed elevation at the inlets of branches $b$ and $c$ are updated by integrating Eq. (8). We then compute the transverse sediment flux $Q_{sy}$ flowing from one node cell to the other according to Eq. (7). Consistently with the discretization of the node cells represented in Fig. 2, at any time step $n$ the average bed elevations of the node cells

$\eta_{bN}$ and $\eta_{cN}$ are computed as

$$\eta_{bN} = \frac{\eta_{a,M-1}^n + \eta_{b,0}^n}{2}, \qquad \eta_{cN} = \frac{\eta_{a,M-1}^n + \eta_{c,0}^n}{2}. \tag{11}$$

Furthermore, the Shields stress $\theta_a$ and solid discharge $Q_{sa}$ corresponding to the incoming flow are retrieved from the last node of the upstream channel $a$. The resulting expression used in the numerical scheme to compute $Q_{sy}$ at the time step $n+1$ reads:

$$Q_{sy}^{n+1} = Q_{sa,M-1}^{n+1}\left(\frac{Q_b^{n+1} - Q_c^{n+1}}{2Q_a} - \frac{\alpha r}{\sqrt{\theta_{a,M-1}^{n+1}}}\frac{\eta_{b,0}^n - \eta_{c,0}^n}{W_a}\right). \tag{12}$$

The update of the bed elevation of a node cell - say, cell $b$ - thus returns:

$$\eta_{b,0}^{n+1} = \eta_{b,0}^n + \eta_{a,M-1}^n - \eta_{a,M-1}^{n+1} - \frac{4\Delta t}{(1-p)\alpha W_a^2}\left(Q_{sb,0}^{n+1} - Q_{sy}^{n+1} - \frac{Q_{sa,M-1}^{n+1}}{2}\right), \tag{13}$$

while an analogous formula is used to update the bed elevation of cell $c$.

## 3  Results

The numerical model described in section 2 is here used to explore the full range of conditions depicted in Fig. 1, for different values of the length of the downstream bifurcates $L$. Each simulation is characterised by a given set of boundary conditions,





namely the water ($Q_0$) and sediment ($Q_{s0}$) discharges supplied to the upstream channel and the water level at the downstream end of the branches $h_d$. We note that when the values of the upstream width $W_a$, grain size $D_s$ and Chézy coefficient $C$ are specified, the water and sediment supply completely determine the reference uniform flow and sediment transport state that

characterises the initial configuration in the three branches, namely the values of the flow depth $D_0$ and the bed slope $S_0$. For a given length $L$ of the bifurcates, these values also fix the downstream water level $h_d$ with respect to the initial bed level at the node.

Simulation results will be presented in dimensionless form, using the parameters of the initial reference state as scaling quantities. As the Chézy coefficient is assumed to be constant, only two parameters are needed to specify in dimensionless

form such reference state, namely its aspect ratio $\beta_0$ and Shields stress $\theta_0$, which are prescribed as input values. Following the lead of BRT, we then quantify the degree of asymmetry between the two bifurcates by means of two dimensionless parameters, namely the discharge asymmetry $\Delta Q$ and the inlet step $\Delta \eta$, that we define as:

$$\Delta Q = \frac{Q_b - Q_c}{Q_0}, \quad \Delta \eta = \frac{\eta_c - \eta_b}{D_0}, \tag{14}$$

where $\eta_b$ and $\eta_c$ stand for the bed elevations at the inlets of the downstream branches. Furthermore, we define the average bed

elevation $\overline{\eta}$ at the inlet of the bifurcates as:

$$\overline{\eta} = \frac{\eta_b + \eta_c}{2D_0}. \tag{15}$$

Finally, we scale the longitudinal coordinate $x$ with the reference flow depth $D_0$ and the time $t$ with the morphodynamic timescale $T_F$ defined by the Exner equation:

$$T_F = \frac{W_a^2 D_0}{Q_{s0}}. \tag{16}$$

As foreseen by the BRT model, in all numerical simulations in which the aspect ratio $\beta_0$ is larger than the critical aspect ratio $\beta_C$ any asymmetric perturbation of the initially balanced bifurcation grows over time. The early stages of these simulations take place according to a well-defined behaviour (later described in this section), which is qualitatively independent of the governing parameters. On the contrary, the long-term state of bifurcations may be of two types depending on the reference conditions of the upstream channel. Specifically, the bifurcation may either reach an equilibrium configuration that corresponds to that

foreseen by the BRT model or evolve towards a considerably more asymmetric type of equilibrium state. The latter type of equilibrium configuration occurs when the transport capacity of the branch carrying the smaller portion of the incoming discharge becomes negligible. We term this condition "partial avulsion".

Whether the bifurcation reaches one type of long-term state or the other depends on the dimensionless parameters of the reference state $\beta_0$ and $\theta_0$. Specifically, we found that the long-term state of numerical simulations depends on which of the

three regions of the $\beta_0 - \theta_0$ space identified by the BRT model (and described in Fig. 1) the provided values of $\beta_0$ and $\theta_0$ belong to. For a given value of $\theta_0$, these regions are identified by the critical aspect ratio $\beta_C$ and the value of $\beta_0$ for which the non-dominant branch has no transport capacity at equilibrium according to the BRT model, i.e., the no-transport aspect ratio $\beta_{NT}$. If $\beta_C < \beta_0 < \beta_{NT}$, the numerical simulation evolves towards the long-term state predicted by BRT. Conversely, when





$\beta_0 \geq \beta_{NT}$ a partial avulsion occurs. In the following, the transient behaviour of these two types of simulations are described

in detail.

Figure 3 shows an example of simulations in which $\beta_C < \beta_0 < \beta_{NT}$, where the system always evolves towards an equilibrium state where both branches actively transport sediments. Since the transport capacity of both branches is non-negligible both during the evolution of the bifurcation and at equilibrium, we will hereafter refer to this category of simulations as "fully active". In this case, the morphodynamic evolution of the bifurcation takes place by means of three distinct evolutionary stages.

In the first stage, the bed elevation gap between the branches - i.e., the "inlet step" defined by BRT - grows over time following an exponential trend. Specifically, while the branch carrying an increasingly larger fraction of the incoming water discharge (the "dominant" branch) undergoes an erosion process, the other branch (the "non-dominant" branch) undergoes a deposition process. As this two opposite processes develop, the bed slopes of the branches don't vary significantly with respect to their initial value $S_0$. On the contrary, the water discharge asymmetry $\Delta Q$ grows exponentially over time (see Fig. 3). The

amplification of the degree of asymmetry of the bifurcation in this stage closely matches that resulting from linear stability analyses of other morphodynamical phenomena, such as alternate bars (Callander, 1969) and dunes and antidunes (Kennedy, 1963).

In the second stage of these simulations, nonlinear effects become dominant as the overall bed elevation and the bed slopes of the branches further evolve. The bed slopes of both branches decrease over time, approaching the equilibrium value foreseen

by the BRT model, while the discharge asymmetry $\Delta Q$ keeps growing until it reaches a maximum value (see Fig. 3).

In the third stage, after the maximum of $\Delta Q$ is reached, the water discharge asymmetry of the bifurcation decreases, showing an upward-concave trend over time, until it reaches its equilibrium value.

It is worth noticing that there is a non-negligible difference between the maximum value of $\Delta Q$ reached at the end of the second stage and its long-term equilibrium value. From now on, we will refer to this difference as "overshoot", as the

simulated value of $\Delta Q$ goes beyond the equilibrium value predicted by the analytical BRT model. The overshoot is due to the morphodynamic evolution of the non-dominant branch: the temporary increase of $\Delta Q$ is in fact caused by an excess of deposition with respect to the final state that occurs at the inlet of the branch. This temporary increase of the bed elevation of the non-dominant branch causes a decrease in the water discharge entering that branch, thus increasing the water discharge asymmetry $\Delta Q$. This phenomenon may play an important role in determining the long-term state of the bifurcation, as will be

discussed later in this chapter.

As a result of these processes, the bifurcation eventually reaches a long-term state that matches the equilibrium configuration foreseen by the analytical BRT model. In this equilibrium state, both branches are characterised by uniform flow, with constant water depth and bed slope over their length. Since the widths of the branches are both equal to half the width of the upstream channel, the equilibrium value of the bed slopes must always be lower than the initial bed slope $S_0$ to ensure the conservation

of sediment mass. Under equilibrium conditions, the inlet step ensures that the transverse sediment flux at the node balances the difference between the transport capacities of the downstream branches, as results from the BRT bulk relationship (7).

When $\beta_0 > \beta_{NT}$, the evolutionary trajectory of the bifurcation doesn't lead it to reach the equilibrium foreseen by the BRT model as it does for fully active simulations. An example of this type of simulations is represented in Fig. 4. By comparing the



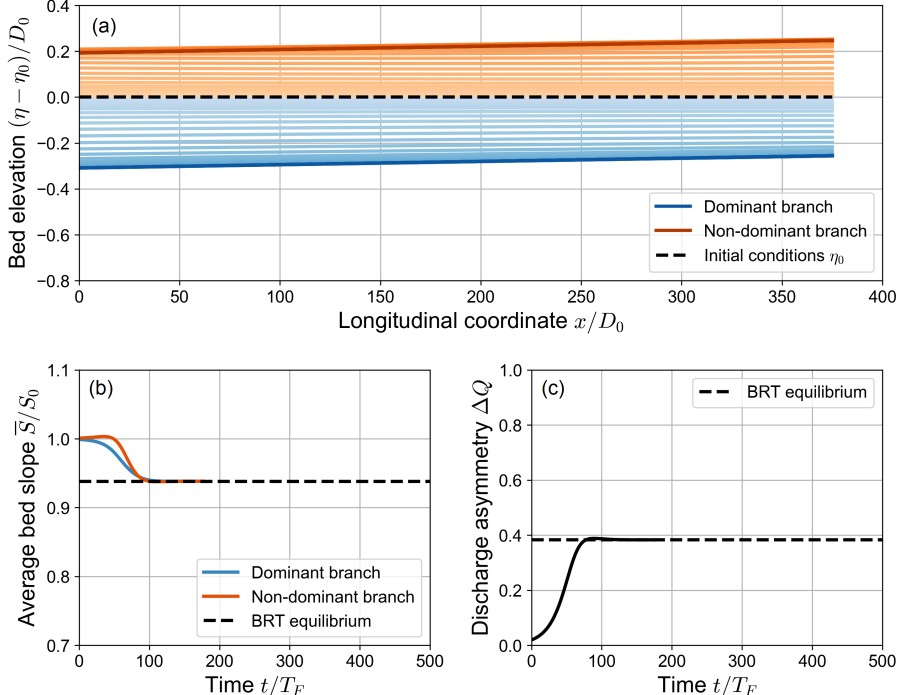

**Figure 3.** Spatial (upper panel) and temporal (lower panels) evolution of a fully active numerical simulation, where the long-term state matches the equilibrium solution of the BRT model ($\beta_0 = 15$, $\theta_0 = 0.07$, $C = 12$, transport formula of Meyer-Peter and Müller (1948)). The upper panel shows the variation over time of the bed elevation profiles with respect to the initial bed profile $\eta_0$ (dashed line). Note that both the longitudinal coordinate $x$ and the bed elevation variation $\eta - \eta_0$ are scaled with the reference water depth $D_0$. The profiles are sampled with a constant time step of $\Delta t/T_F = 4.4$, where $T_F$ is defined as in Eq. (16). The lower left panel shows the evolution of the average bed slope of the branches, scaled with the initial bed slope $S_0$. The lower right panel shows the evolution over time of the water discharge asymmetry $\Delta Q$ (defined as in (14)). In both lower panels, the dashed line indicates the equilibrium solution foreseen by the BRT model.

evolution over time of the discharge asymmetry $\Delta Q$ of this type of simulations (panel (c) of Fig. 4) with that of fully active
simulations (panel (c) of Fig. 3), one can see that their first two stages qualitatively resemble each other. However, contrary
to fully active simulations, at the end of the second stage of simulations where $\beta_0 > \beta_{NT}$ the transport capacity of the non-
dominant branch becomes negligible. As a result, the dominant branch is fed with all the incoming sediment discharge and,
most importantly, the non-dominant branch is not able to adjust its bed slope over time anymore. After this "freezing" of the
bed slope of the non-dominant branch, the water discharge and the average water depth of the dominant branch considerably
increase over time, while its bed slope decreases. This leads to an overall degradation of the branch. Since the non-dominant
branch is no longer capable of modifying its morphology, this degradation sharply increases the difference in water depth
between the branches, as the water level is prescribed to be the same at the inlets of the branches. The discharge asymmetry



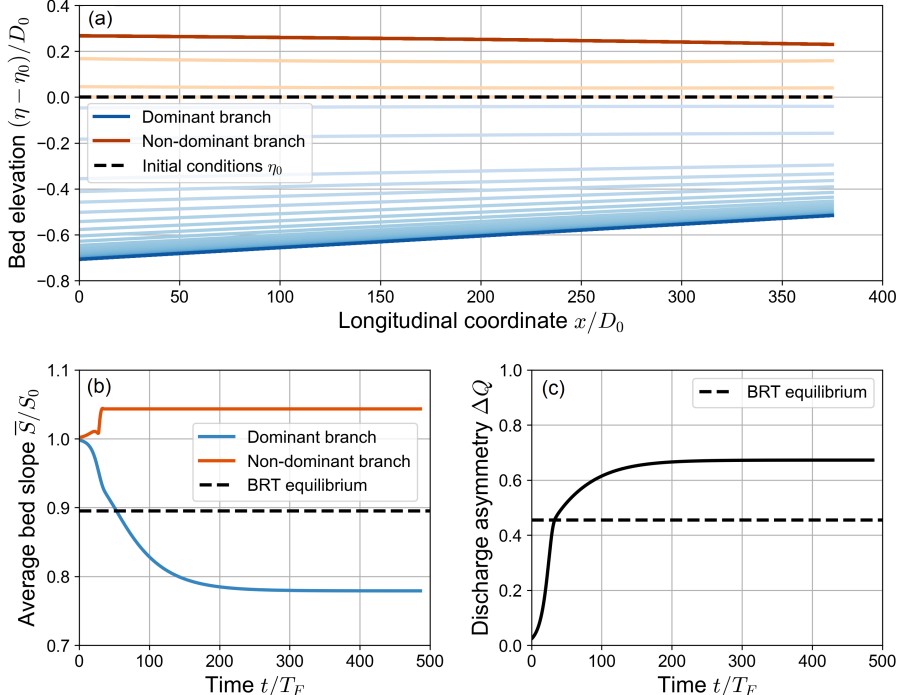

**Figure 4.** Spatial (upper panel) and temporal (lower panels) evolution of a numerical simulation where a partial avulsion occurs ($\beta_0 = 20$, $\theta_0 = 0.07$, $C = 12$, transport formula of Meyer-Peter and Müller (1948)). The upper panel shows the variation over time of the bed elevation profiles with respect to the initial bed profile $\eta_0$ (dashed line). Note that both the longitudinal coordinate $x$ and the bed elevation variation $\eta - \eta_0$ are scaled with the reference water depth $D_0$. The profiles are sampled with a constant dimensionless time step of $\Delta t/T_F = 12.2$, where $T_F$ is defined as in Eq. (16). The lower left panel shows the evolution of the average bed slope of the branches, scaled with the initial bed slope $S_0$. The lower right panel shows the evolution over time of the water discharge asymmetry $\Delta Q$ (defined as in (14)). In both lower panels, the dashed line indicates the equilibrium solution foreseen by the BRT model.

$\Delta Q$ keeps increasing as well, asymptotically reaching a much larger equilibrium value than that foreseen by the BRT model (see Fig. 4).

At equilibrium, the bed slope of the dominant branch is significantly lower than the equilibrium slope predicted by BRT. On the contrary, the bed slope of the non-dominant branch remains close to the slope $S_0$ prescribed as initial condition. This is due to the fact that, as in the fully active simulations, the bed slope of both branches doesn't vary significantly in the early evolutionary stages, while the inlet elevation gap grows over time. The equilibrium configurations of the two bifurcates also differ in terms of water-surface profile. While the dominant branch is characterised by uniform flow conditions, the water

surface along the non-dominant branch follows an upward-concave profile. Being the transport capacity equal to zero along this branch, the backwater curve doesn't imply an ongoing morphodynamic evolution, and is therefore compatible with an equilibrium condition.





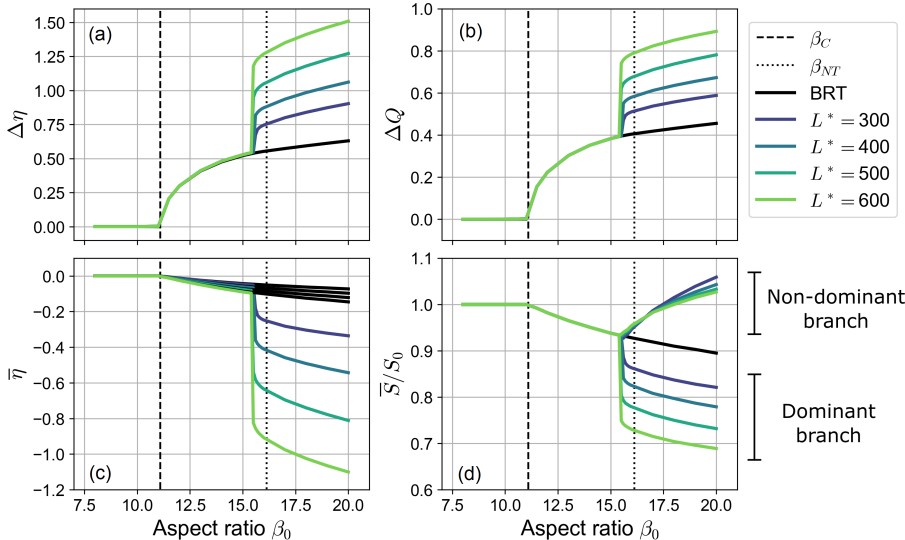

**Figure 5.** Long-term equilibrium states of a free bifurcation: the four panels show the dependence on the aspect ratio $\beta_0$ of the inlet step $\Delta\eta$, the water discharge asymmetry $\Delta Q$, the average bed elevation at the inlets of the branches $\overline{\eta}$ and of the bed slope averaged over the branch $\overline{S}$, for different values of the non-dimensional length of the bifurcates $L^* = L/D_0$ ($\theta_0 = 0.07, C = 12$, transport formula of Meyer-Peter and Müller (1948)). Solid black lines indicate the equilibrium solutions computed by means of the BRT model. Dashed and dotted lines represent the critical aspect ratio $\beta_C$ and the no-transport aspect ratio $\beta_{NT}$, respectively.

It is worth noticing that the two types of equilibrium configurations described above show a remarkably different dependency on the governing parameters. As shown in Fig. 5, the equilibrium configuration of a fully active bifurcation (i.e. when $\beta_C <$ 290 $\beta_0 < \beta_{NT}$) doesn't depend on the length of the bifurcates, as foreseen by the BRT model, the resulting asymmetry being mainly governed by the parameters of the reference upstream flow (i.e., the aspect ratio $\beta_0$). On the contrary, when a partial avulsion occurs the final configuration also exhibits a sharp dependence on the length of the downstream branches. Specifically, bifurcations with longer branches show a larger discharge asymmetry, a lower bed slope of the dominant branch and – as a result – a larger overall degradation of the dominant branch itself, as shown by the considerably lower values of $\overline{\eta}$.

The scenario depicted in Fig. 5 does not qualitatively change when changing the reference Shields stress $\theta_0$ but for the values of the critical and the no-transport aspect ratio, whose dependence on $\theta_0$ is illustrated in Fig. 1.

The results of numerical simulations therefore suggest that in the case of a partial avulsion the equilibrium state gets more unbalanced as the dimensionless length of the bifurcates $L^* = L/D_0$ increases. Interestingly, by further increasing the length of the branches the evolution of the bifurcation eventually leads to the complete abandonment of the non-dominant branch. In 300 these cases, which we call "full avulsions", the dominant branch carries all the incoming water and sediment discharges.

Figure 5 also shows that the exact value of $\beta_0$ above which partial avulsions occur in the numerical simulations is slightly lower than the threshold value $\beta_{NT}$ computed analytically (see dotted lines in the figure). This offset is due to the overshoot phenomenon: if the temporary decrease in the water discharge flowing in the non-dominant branch is large enough, the transport





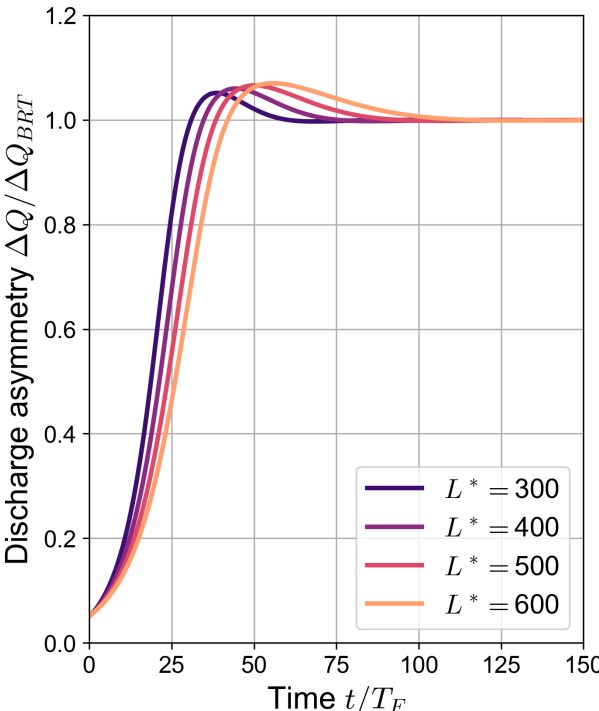

**Figure 6.** The variation over time of the ratio between the simulated discharge asymmetry $\Delta Q$ and the equilibrium value foreseen by the BRT model $\Delta Q_{BRT}$, for different values of the dimensionless length of the bifurcates $L^* = L/D_0$ ($\beta_0 = 18$, $\theta_0 = 0.07$, $C = 12$, transport formula of Parker (1978)). The time $t$ is made dimensionless by means of the Exner timescale $T_F$, defined as in Eq. (16).

capacity of the branch may drop to zero, thus causing a partial avulsion. Overall, the dependence of the amplitude of the
overshoot on the length of the bifurcates is rather weak, as shown by Fig. 6. Thus, the threshold for $\beta_0$ for which a partial avulsion occurs in the numerical simulations is almost independent of the length of downstream branches.

Based on the results of numerical simulations, we can therefore identify different categories of the long-term equilibrium state of a free bifurcation in the $\beta_0 - L^*$ plane, as reported in Fig. 7. When $\beta_0$ is lower than the critical aspect ratio $\beta_C$, all stable equilibrium states of the bifurcation are balanced, i.e., show an even partitioning of water and solid discharge. Instead,
when $\beta_0 > \beta_C$ the bifurcation evolves towards a stable and unbalanced equilibrium state, where both branches effectively carry sediments ("fully active" simulations). When $\beta_0$ exceeds a second threshold $\beta_{NT}$, which is found numerically to be slightly lower than the no-transport threshold predicted analytically, a partial avulsion occurs during the evolution of the bifurcation. In this case, the equilibrium state can be of two types, depending on the value of the dimensionless length of the bifurcates $L^*$. For lower values of $L^*$, the non-transporting branch remains open, although the water discharge asymmetry is much larger than
that resulting in the fully active region and also depends on the length of the branches, while it keeps a similar dependency on $\beta_0$. At larger $L^*$ values, a full avulsion occurs and the non-dominant branch is completely abandoned. As suggested by Fig. 7



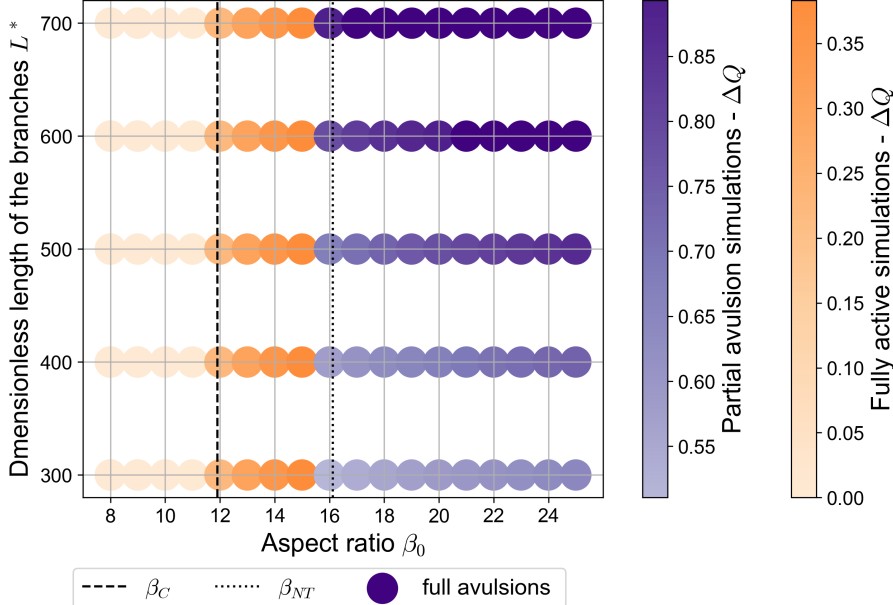

**Figure 7.** Equilibrium configurations of a free bifurcation for different values of the dimensionless length of the downstream branches $L^*$ and of the aspect ratio $\beta_0$ ($\theta_0 = 0.07$, $C = 12$, transport formula of Meyer-Peter and Müller (1948)). Each point corresponds to a single numerical simulation. Reddish colours indicate fully active states, where the discharge asymmetry $\Delta Q$ predicted by BRT is attained. Bluish tones indicate partial avulsion states, in which one of the bifurcates does not transport sediments, while dark blue dots indicate states where one of the branches in completely abandoned ($\Delta Q \simeq 1$, full avulsions). Note that the value of the reference Shields stress is the same for all simulations.

the "avulsion length" $L^*_{AV}$, i.e. the threshold value of $L^*$ for which a full avulsion occurs, is weakly dependent on the upstream aspect ratio $\beta_0$.

## 4 Simplified equilibrium model for partial avulsions

The results of numerical simulations reported in the previous section suggest that under conditions of partial avulsion the long-term equilibrium state of a free evolving bifurcation significantly differs from that predicted by the analytical BRT model. Nonetheless, in this section we show how the equilibrium value of the discharge asymmetry can be still predicted by a simple, analytical model by introducing appropriate hypotheses that are supported by information retrieved from numerical simulations.

We consider the long-term state of an initially balanced bifurcation depicted in Fig. 8, where the shoaling branch $c$ is not
able to effectively transport sediments. As in the BRT model, the upstream water ($Q_0$) and sediment ($Q_{s0}$) supplies are given, while the water level at the downstream end of the branches $h_d$ is fixed and defined by the reference water depth $D_0$. However, in this case two additional pieces of information are required to close the mathematical formulation of the equilibrium state, namely the slope $S_c$ and the inlet bed elevation $\eta_c$ of the shoaling branch. In fact, their actual value result from the evolutionary



process of the bifurcation, being determined when the transport capacity vanishes. Moreover, the sediment transport partition
at the node takes a simple form, as all the incoming solid discharge is conveyed by the dominant branch.

Under fully active conditions, both downstream branches evolve towards a uniform flow and sediment transport config-
uration, in which the water surface and bed elevation profiles are straight and parallel to each other. This implies that the
equilibrium bed slopes are the same ($S_b = S_c$), as assumed by BRT. However, this is no longer the case under partial avulsion
conditions, because when the transport capacity of the shoaling branch vanishes its bed slope $S_c$ remains constant over time,
while the slope $S_b$ of the dominant branch continues to evolve until equilibrium is attained. By looking closely at the transient
behaviour of the shoaling branch displayed by numerical simulations, we may observe that the overall homogeneous deposi-
tion occurring along the branch causes it to stop transporting sediments before its bed slope changes appreciably. Therefore,
numerical results suggest that the slope of the shoaling branch does not vary significantly with respect to the initial condition,
so that it is reasonable to assume that, at equilibrium:

$$S_c = S_0. \tag{17}$$

As can be seen in panel (d) of Fig. 5, the error we make by using Eq. (17) is relatively small, namely of order $10^{-2} S_0$.

Another key difference of partial avulsions with respect to the fully-active configurations is the computation of the transverse
flux of sediments at the node $Q_{sy}$. Under fully-active conditions $Q_{sy}$ is computed by means of the BRT bulk relationship (7),
and depends on both the discharge asymmetry and the bed elevation gap between the branches. Under conditions of partial
avulsion, the equilibrium value of the transverse flux of sediment at the node is always equal to half the incoming sediment
discharge, thus granting solid mass continuity in the node cells, regardless of the bed elevation gap between the branches.
Thus, the transverse bed slope adjusts in order to satisfy the relationship between the discharge asymmetry and $Q_{sy}$ defined
by Eq. (7). On the other hand, the bed elevation gap between the downstream branches – i.e., the inlet step $\Delta\eta$, as defined
in Eq. (14) – is entirely governed by the degradation of the dominant branch that leads it to reach its equilibrium bed slope.
Thus, the node cells and the inlet step evolve according to two different mechanisms, and do not coincide anymore as they do
in fully-active conditions. As a result, the system gains a degree of freedom.

To replace the missing condition, we assume that, when the transport capacity of the non-dominant branch vanishes, the inlet
step $\Delta\eta$ equals the inlet step $\Delta\eta_{BRT}$ foreseen by the BRT model. This assumption is consistent with the numerical results,
since the first stages of partial avulsion and fully active simulations evolve in a very similar manner. This allows us to compute
the bed elevation $\eta_c$ at the inlet of the shoaling channel $c$, defined with respect to the initial bed elevation, as

$$\eta_c = \frac{\Delta\eta_{BRT}}{2} D_0. \tag{18}$$

We also note that the variations of discharge asymmetry after the freezing of the shoaling branch imply that flow conditions
do not keep uniform throughout the channel. Specifically, the reduction of the water discharge $Q_c$ and the fixed water surface
elevation at the downstream end of both branches produce an upward concave water-surface profile in the shoaling branch, as
illustrated in Fig. 8. Nonetheless, when the branches are longer than the backwater length, it is still possible to assume that in
the upstream part of the shoaling branch the flow keeps nearly uniform. This allows us to consider the following uniform flow





relations for both branches:

$$Q_b = W_b C \sqrt{gS_b} D_b^{3/2}, \tag{19a}$$

$$Q_c = W_c C \sqrt{gS_c} D_c^{3/2}, \tag{19b}$$

where the water fluxes $Q_b$ and $Q_c$ can be expressed as functions of the discharge asymmetry $\Delta Q$ by considering the water mass conservation:

$$\frac{Q_b}{Q_0} = \frac{1 + \Delta Q}{2}, \quad \frac{Q_c}{Q_0} = \frac{1 - \Delta Q}{2}. \tag{20}$$

By substituting Eq. (18) into the nodal condition that sets the same water surface elevation at the inlets of branches $b$ and $c$, we obtain:

$$D_b - D_c = \eta_c - \eta_b = \frac{\Delta \eta_{BRT}}{2} D_0 - \eta_b, \tag{21}$$

where the bed elevation at the inlet of the dominant branch $\eta_b$ is given by the following geometrical relation, illustrated by Fig. 8:

$$\eta_b = -(S_0 - S_b)L - (D_b - D_0). \tag{22}$$

The system of equations is finally completed by the sediment mass conservation, which under partial avulsion conditions

becomes particularly simple, as the entire sediment supply is carried by the dominant branch $b$:

$$\Phi(\theta_b) = \Phi(\theta_0)\frac{W_a}{W_b} = 2\Phi(\theta_0), \tag{23}$$

where the Shields stress $\theta_b$ takes the following form:

$$\theta_b = \frac{S_b D_b}{\Delta D_s}. \tag{24}$$

Summarising, the analytical model for the the long-term state of bifurcations under partial avulsion conditions can be cast

into a system of four algebraic equations in terms of the four unknowns $S_b$, $D_b$, $D_c$ and $\Delta Q$. Specifically, by substituting (19) into (20), (22) into (21) and using (23, 24) we obtain:

$$\begin{cases} \dfrac{D_c}{D_0} + \dfrac{\Delta \eta_{BRT}}{2} = 1 - (S_0 - S_b)\dfrac{L}{D_0}, \\ \sqrt{\dfrac{S_b}{S_0}} \left(\dfrac{D_b}{D_0}\right)^{3/2} = 1 + \Delta Q, \\ \left(\dfrac{D_c}{D_0}\right)^{3/2} = 1 - \Delta Q, \\ \Phi(\theta_b) = 2\Phi(\theta_0). \end{cases} \tag{25}$$

Despite its approximate nature, our simple analytical model proves capable of capturing the key ingredients that determine the equilibrium configurations reached by numerical simulations in which a partial avulsion occurs. As shown by Fig. 9, the



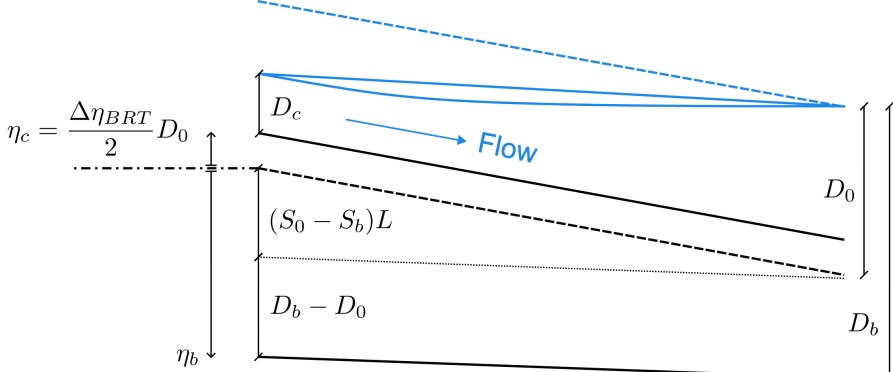

**Figure 8.** Schematic view of two anabranches that undergo a partial avulsion. The bed elevation and water surface profiles are shown for both the initial condition (dashed lines) and long-term configuration (solid lines). $\eta$ stands for the bed elevation at the inlet of the bifurcates, while $S$ is the longitudinal bed slope, $L$ is the length of both branches and $D$ is the water depth. The subscripts $b$ and $c$ refer to the dominant and non-dominant branch, respectively, while the subscript $0$ denotes the initial reference conditions, which correspond to the initial conditions of the downstream branches. The dash-dotted line indicates the reference level for the bed elevation $\eta$, while the dotted line is plotted to highlight the two contributions to the bed elevation $\eta_b$, as illustrated by Eq. (22).

equilibrium discharge asymmetry $\Delta Q$ foreseen by the analytical model closely matches that obtained in numerical simulations, with an average relative accuracy of less than 3%. Specifically, our simplified model is able to correctly reproduce the dependence of $\Delta Q$ on both the aspect ratio $\beta_0$ and the dimensionless channel length $L^*$. On the contrary, the equilibrium values of $\Delta Q$ computed using the BRT model, besides being independent on the length of the bifurcates, are considerably lower than those shown by numerical simulations, although the dependence of $\Delta Q$ on the aspect ratio $\beta_0$ is similar.

## 5    Discussion

The formulation of a new evolutionary model for a free bifurcation allowed us to study the equilibrium states in the so-called partial avulsion conditions, in which the flow velocity in one of the bifurcates is not sufficient to mobilise the bed material. In these conditions, the resulting equilibrium configuration turns out to be very different from that predicted by the Bolla Pittaluga et al. (2003) model. Specifically, the resulting long-term equilibrium solution is considerably more asymmetrical, and the inlet

step is significantly higher. This happens because the bed slope of the shoaling branch is no longer able to adjust when sediment transport vanishes, while the dominant branch unavoidably undergoes a bed degradation process, which enhances both the inlet step and the discharge asymmetry. Bed degradation increases with the length of the branches, which therefore becomes a key control for the equilibrium configuration.

     Consistently with Fig. 1, partial avulsions occur when the aspect ratio of the upstream channel exceeds the threshold value

$\beta_{NT}$. However, the actual value of the threshold displayed by simulations turns out to be slightly lower due to an overshoot phenomenon, i.e. in the transitory phase the discharge asymmetry reaches a value that exceeds the long-term equilibrium value.





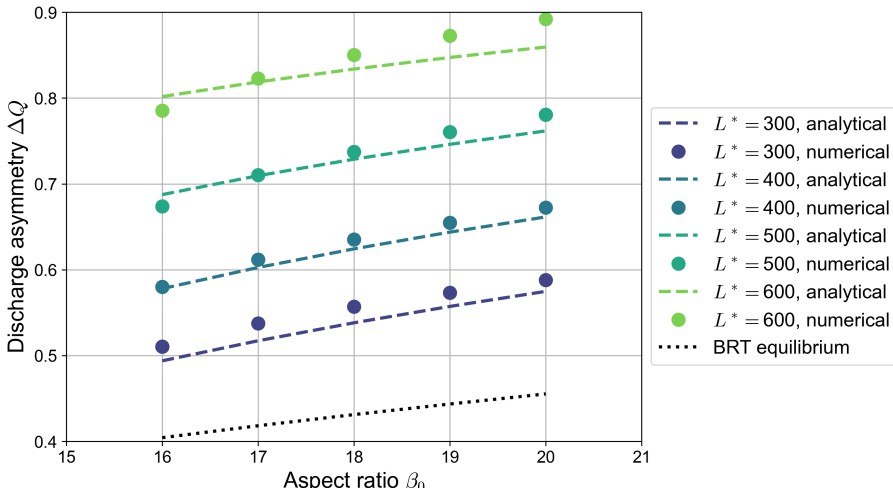

**Figure 9.** Comparison between the equilibrium values of the discharge asymmetry $\Delta Q$ of bifurcations undergoing partial avulsion as computed by numerical simulations (dots) and those predicted by the analytical model (dashed lines), for different values of the dimensionless length of the bifurcates $L^* = L/D_0$ ($\theta_0 = 0.07, C = 12$, transport formula of Meyer-Peter and Müller (1948)). Each dot represents the outcome of a single numerical simulation. The dotted line indicates the equilibrium solution computed by means of the BRT model, which is independent of the length of the branches.

Moreover, the actual threshold for numerical simulations to undergo a partial avulsion may be further reduced with respect to the theoretical value $\beta_{NT}$, depending on the behaviour of the transport formula near critical conditions. For example, when considering the transport formula of Parker (1978), the bed evolution of the non-dominant branch "freezes" when the Shields stress is only slightly above the critical threshold $\theta_{cr} = 0.03$. This occurs because what actually determines the transition from a fully-active condition to partial avulsion is the ratio between the evolutionary timescales of the branches. Specifically, if the dominant branch degrades too quickly with respect to the other branch, it starts capturing more discharge than that foreseen by the BRT model, which rapidly leads to partial avulsion conditions. It is worth noting that, for the same reason, partial avulsions also occur when employing a transport formula that does not consider a critical threshold (e.g., Parker, 1990). In this case, the analytical threshold $\beta_{NT}$ beyond which the BRT model doesn't work can no longer be defined. However, numerical simulations still show a threshold for $\beta_0$ above which partial avulsions occur. In these conditions, the shoaling branch practically freezes even though a minimal amount of sediment transport is still present, causing the bifurcation to reach a strongly unbalanced long-term configuration.

The analytical model developed in Sect. 4 enables for computing the long-term state of bifurcations under conditions of partial avulsion, and for identifying the key parameters that control the transition between the different kinds of equilibrium configurations shown by numerical simulations and represented in Fig. 7. When combined with the results of the BRT model, the analytical model allows us to draw a complete picture of the nature of the equilibrium solutions in the $\beta_0 - L^*$ parameters



space, as illustrated in Fig. 10. Specifically, for a given value of the reference Shields stress $\theta_0$, four distinct regions are identified:

1. balanced solutions (i.e., $\Delta Q = 1/2$), occurring when the aspect ratio $\beta_0$ is lower than the length-independent critical threshold $\beta_C$;

2. unbalanced fully-active solutions, where at equilibrium both branches do actively transport sediment, occurring when $\beta_0$ lies between $\beta_C$ and the no-transport threshold $\beta_{NT}$;

3. partial avulsion solutions, in which all sediments go in one branch, while a nonzero water discharge flows in the other,
occurring when $\beta_0 > \beta_{NT}$ and the length of the branches is shorter than the threshold $L_{AV}^*$;

4. full avulsion solutions, where one branch is abandoned and all the incoming water and sediment supplies are carried by the other branch, occurring when the aspect ratio is relatively large and the downstream branches are relatively long (i.e., $\beta_0 > \beta_{NT}$ and $L^* > L_{AV}^*$).

The threshold values $\beta_C$ and $\beta_{NT}$ increase as the Shields stress increases, and so does the avulsion length $L_{AV}^*$, while it
exhibits a weak dependence on $\beta_0$.

As shown in Fig. 10, when the downstream branches are sufficiently long the degradation process can eventually lead to the complete abandonment of the shoaling branch, so that water and sediment fluxes are entirely conveyed in a single thread (full avulsion). The dimensionless branch length at which full avulsion occurs ($L_{AV}^*$) is found to scale with the backwater length. This result is consistent with existing studies of river deltas that show how the distance of apex avulsions from the coastline
scales with the backwater length (Chatanantavet et al., 2012; Jerolmack and Mohrig, 2007). It is worth remarking that previous studies found this scaling by analysing deposition induced by time-dependent flow discharge, while our study highlights the link between the backwater length and the threshold for avulsions by analysing the interplay between the downstream boundary condition and the incision required for the abandonment of the shoaling channel.

Our analysis provides a clear example of how the complete abandonment of one of the two branches can be caused by
the incision of the dominant channel driven by an autogenic mechanism. In general, bed incision takes place whenever the bifurcation becomes unbalanced, as the resulting difference of the Shields stress between the two branches makes them overall more efficient in transporting sediments (Paola, 1996; Ferguson, 2003). Therefore, a smaller slope is sufficient to ensure equilibrium with the imposed sediment supply. This effect is further amplified when the water discharge asymmetry sharply increases due to a partial avulsion. This can provide a mechanistic interpretation of the so-called process of incisional avulsion
(Mohrig et al., 2000; Slingerland and Smith, 2004), in which channel abandonment is preceded by headward erosion of the avulsed channel.

The tendency to abandon smaller branches can be seen as a general characteristic of rivers systems undergoing significant incision, as that produced by a lack of sediment supply. Specifically, discharge asymmetry of river bifurcations generally increases when the overall degradation rate is relatively rapid. In this case, the shoaling branch is not able to keep up with the
erosion, thus becoming increasingly superelevated with respect to the dominant channel. This may provide a way to interpret





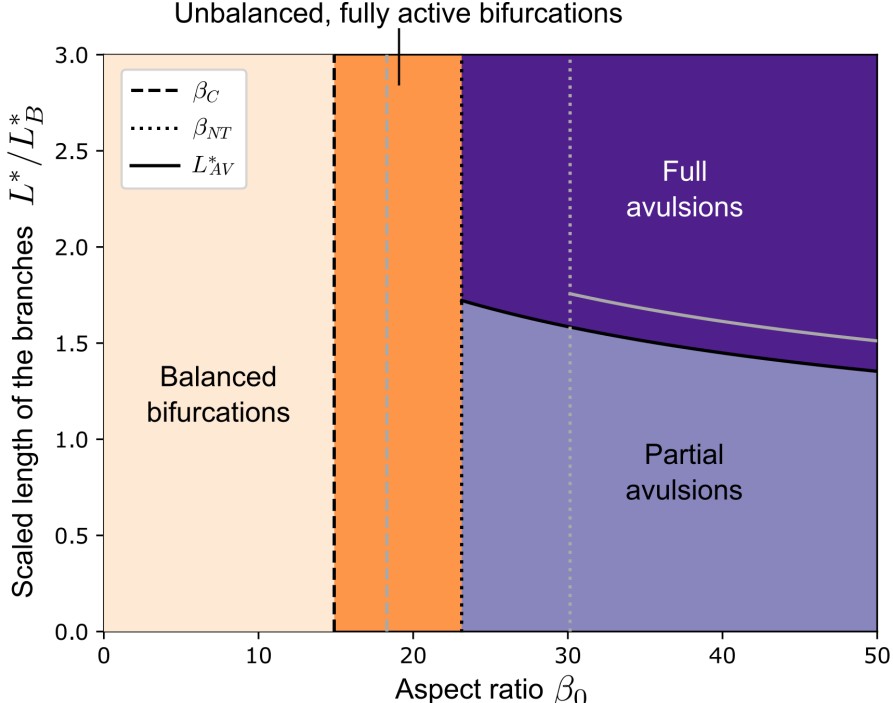

**Figure 10.** The different types of equilibrium configurations of a free bifurcation in the $\beta_0$ (aspect ratio) - $L^*$ (dimensionless branch length) parameter space, resulting from the combination of the BRT model and our analytical model ($\theta_0 = 0.08$). The length of the bifurcates is scaled with the dimensionless backwater length $L_B^*$. The different line styles indicate the critical aspect ratio $\beta_C$, the aspect ratio of no-transport $\beta_{NT}$ and the dimensionless full avulsion length $L_{AV}^*$. The grey lines show how these thresholds change when considering a slightly higher value of the Shields stress ($\theta_0 = 0.09$).

the morphological evolution of braided rivers under conditions of reduced sediment availability, which typically lead to the simplification of the channel network and eventually to the transition to single-thread configurations (e.g., Germanoski and a. Schumm, 1993; Marti and Bezzola, 2006; Redolfi and Tubino, 2014).

The numerical and analytical results reported in Fig. 5 and Fig. 9 highlight the inadequacy of the BRT model to reproduce
the strongly unbalanced long-term configuration of free bifurcations when a branch avulsion occurs, as already pointed out by Kleinhans et al. (2008). They also provide a new insight into the long-debated question on the role of the nodal condition for sediment partitioning. In fully active conditions, the inlet step between the downstream branches of an unbalanced bifurcation generates a lateral bed slope at the node, such that the transverse sediment flux $Q_{sy}$ compensates the imbalance of transport capacities between the bifurcates themselves. The efficiency of this lateral deflection thus plays a key role in determining the
equilibrium configuration of the bifurcation, as shown by the BRT model. On the contrary, under partial (or full) avulsion conditions the incoming sediment supply is entirely steered towards the dominant branch, regardless of the inlet step. After the shoaling branch gets morphodynamically inactive, the development of the inlet step between the bifurcates is entirely governed




by the evolution of the dominant branch, and specifically by the changes of its bed slope and water depth that are necessary to attain a morphodynamic equilibrium.

Under this condition, the transverse flux of sediments $Q_{sy}$ is mainly driven by flow exchange (the first term on the left-hand side of the nodal relationship (7)), while the lateral deflection due to transverse slope (the second term on the left-hand side of Eq. (7)) plays a minor role in determining the equilibrium bifurcation asymmetry. As a result, the bed elevation gap between the node cells of the BRT model (see Fig. 2) simply yields the residual contribution ensuring the condition $Q_{sy} = Q_{s0}/2$ at equilibrium, and it does no longer align with the downstream inlet step. As can be seen in Fig. 4, the resulting evolution of

the dominant branch shows no discontinuity between the fully-active initial stages and the partial-avulsion long-term state. In fact, the transverse flux returned by Eq. (7) produces a smooth transient behaviour in the dominant channel, and ensures mass conservation.

In general, the timescale of the adaptation of the bifurcation to its long-term state is much longer than that governing the first evolutionary stages, as shown for example by the lower-right panel of Fig. 4. This happens because the dominant branch needs

time to adapt its bed slope to the equilibrium value. In the meanwhile, other upstream or downstream effects may influence the evolutionary process of the bifurcation, namely the effect of changes in sediment supply, the adaptation of channel width, variations of the downstream water level (e.g., sea level rise). While our model can be straightforwardly extended to reproduce such effects, further investigation is needed to asses their relevance in determining the bifurcation configuration, depending on the characteristic timescales of the associated processes.

Finally, it is worth noting that in this work we consider the water discharge to be constant in time. Therefore, the application of the model to unsteady flow conditions is only possible when flow variations are sufficiently slow to be represented as a sequence of quasi-steady states. To account for flow variability it would be possible to define an equivalent, formative value of $Q$, for example by considering the method of the effective discharge proposed by Wolman and Miller (1960), or to adopt a procedure similar to that recently proposed by Carlin et al. (2021) for alternate bars.

## 6 Conclusions

In this study we developed a novel 1-D numerical model for predicting the evolution of a river bifurcation. Based on numerical findings, we then built a new analytical model, specifically designed to reproduce the equilibrium configuration of a bifurcation under partial avulsion conditions. Coupling the latter model with that proposed by Bolla Pittaluga et al. (2003) provides a suite of analytical tools that covers the full range of possible long-term states of a free bifurcation in the controlling parameters

space, namely the aspect ratio and the Shields stress.

The combined analysis of the numerical and analytical model yields the following conclusions:

- When the aspect ratio of the main channel exceeds the threshold value $\beta_{NT}$, the bifurcation undergoes partial avulsion conditions, in which the shoaling channel is no longer able to effectively transport sediments.

- Under partial avulsion conditions, the dominant channel is subject to degradation, while the morphological evolution of

the other branch is inhibited. This produces a significant increase of the inlet step at the entrance of the bifurcates, further



enhancing the water discharge entering the dominant channel. This autogenic positive feedback mechanism results in a highly unbalanced equilibrium configuration, whose long-term degree of asymmetry is influenced by the length of the downstream branches.

- When the dimensionless length of the bifurcates exceeds a threshold value $L_{AV}^*$, the discharge asymmetry tends to unity, which implies the complete abandonment of the shoaling channel (full avulsion). Such threshold length is found to scale with the backwater length.

Overall, this work allows us to draw for the first time a complete picture of conditions that may lead to the abandonment of one branch in "free" river bifurcations. This provides the basis for more detailed studies about the role of other factors – such as flow variability, sea level rise and bank erosion – in determining the long-term evolution of bifurcations. Furthermore, our study gives valuable insights for the design of sustainable river restoration interventions aimed at promoting the formation of rich, multi-thread channel patterns.

*Code availability.* The code used for the numerical model will be made freely available upon manuscript acceptance.

*Author contributions.* GB, MR and MT conceptualized the work and designed the methodology. GB developed the software and wrote the manuscript draft, which was reviewed and edited by all authors.

*Competing interests.* The authors declare that they have no conflict of interest.





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
