# Peer review of "Analysis of autogenic bifurcation processes resulting in river avulsion"

_EGUsphere, 2023_

## Author Comment (AC2)

We thank the Reviewer for the constructive comments and suggestions. In the following we provide a detailed response to each point. The Reviewer's comments are in blue text, while our replies are black.

This work investigates the control of avulsions in bifurcated channel systems; a main channel splitting into 2 equal sub channels. Towards this end, the authors build a mathematical/numerical model, extending the well known analysis presented by Bolla Pittaluga et al. (2003) (BRT). The proposed model can track the evolution of a perturbation at the branching point of the sub-channels. In the first place, the model recovers the regimes identified in BRT. At low channel aspect ratios, following a perturbation, the system recovers balanced flows in each of the sub-channels. Beyond a critical aspect ratio Beta_c, however, the long term equilibrium of the flow in the system is unbalanced. This work show that as the aspect ratio is increased further, a second threshold Beta_TH is reached, here the sub channel, with the least flow, exhibits a partial-avulsion—where the channel still carries flow but ceases to transport sediment. Depending on the length of the sub channels, as the aspect ratio is increased even further, a point is reached where full avulsion occurs, i.e. the channel does not convey either water or sediment.

1. I think that the analysis in this paper is of sufficient interest to stand alone but also feel that it would be significantly enhanced, if the authors can point toward experimental or field evidence of the behaviors predicted by the math. For example, referring to Figure (9), it appears, by my calculations, that full avulsion would be reached in a system with channels of water depth of 2 m, width 40 m, and length 1 km+. How common are such conditions in field settings? (eg. Wax Lake in Louisiana)? Are records of permanently avulsed channels seen in such systems? Are records of permanently avulsed channels seen in field systems with shorter channel lengths and smaller aspect ratios?

We thank the Reviewer for the suggestions. Unfortunately, the available datasets are not sufficient to allow a detailed verification of model results.

Data from physical models are scarce, and experiments were mainly focused on the analysis of the effect of specific "forcing" factors. For example, the laboratory experiments by Salter et al. (2019) were tailored to depositional environments and reproduced bifurcations with prograding branches, while Szewczyk et al. (2020) analysed the influence on water discharge partition of changing the bifurcation angle, for values of the aspect ratio $\beta_0$ much smaller than

[Figure]

Figure AC1: Results from the analysis of experimental data by Bertoldi and Tubino (2007). Consistently with our theory, experiments where full avulsions were observed (filled markers) fall in the region $\beta_0 > \beta_{NT}$ and $L > L_{AV}$.

the no-transport threshold $\beta_{NT}$. To the authors' knowledge, the laboratory experiments by Bertoldi and Tubino (2007) are the only ones that analyse how initially balanced "free" bifurcations develop unbalanced configurations, possibly leading to partial or full avulsion. Indeed, a specific analysis of their results seem to support our findings, as shown in Figure AC1 where we report, for each experimental run, the computed values of $\beta_0/\beta_{NT}$ and $L/L_{AV}$. Consistently with our predictions, they reveal that full avulsions (i.e. the complete closure of one of the branches) were observed in the three experimental runs in which $\beta_0$ was greater than (or closer to) $\beta_{NT}$ and the channel length $L$ was longer than the avulsion length $L_{AV}$.

Further support to our findings is also provided by the results of Ragno et al. (2022) who analysed nearly 200 bifurcation-confluence units of sand-bed and gravel-bed rivers. They found that field data show the existence of quasi-universal relations for the branches length when scaled with bankfull variables of the main upstream channel (the mean depth or width). The resulting average dimen-

[Figure]

Figure AC2: Dependence of (a) the dimensionless avulsion length $L_{AV}/D_0$ and (b) the scaled avulsion length $L_{AV}/L_B$ on the Shields stress $\theta_0$, as obtained from Equation (AC4). Vertical dashed lines indicate the value of Shields stress for which $\beta_0 = \beta_{NT}$, thus representing the boundary of the region of validity of the analytical model. The dotted line indicates the upper bound for $L_{AV}$, obtained by neglecting the $\beta$-dependent term $\Delta\eta_{BRT}$ ($C = 12$, transport formula by Meyer-Peter and Müller, 1948).

sionless length falls in the range $L^* = 200 - 300$, which can be interpreted as a preferential range of length values that allows river loops to keep both branches active. The results reported in Figure AC2a suggest that these values are typically lower than the predicted dimensionless avulsion length $L_{AV}/D_0$. We note, however, that the application of our model to this data-set is not straightforward, as the presence of the confluence may have an important influence on loop stability and equilibrium configuration (Ragno et al., 2021). Therefore, a direct comparison with our model's results would require incorporating the effect of the confluence, and considering the peculiar hydrodynamical and morphodynamical characteristics of individual field cases.

Regarding the estimate of the avulsion length proposed by the Reviewer based on Figure 9 of our paper, we observe that the figure does not include conditions of full avulsion, which would correspond to the intersection of the curves with the $\Delta Q = 1$ line (incidentally, we also note that we define the aspect ratio $\beta_0$ as the half-width to depth ratio). To make this point clearer, we hereby provide the mathematical procedure, based on the analytical model described in Section 4 of our work, to compute the avulsion length $L_{AV}$ as a function of the flow conditions in the upstream channel, namely the Shields stress $\theta_0$ and the channel aspect ratio $\beta_0$, provided the latter is larger than the no-transport aspect ratio $\beta_{NT}$. We'll also add the following paragraphs in the revised version of the manuscript.

The analytical model makes it possible to determine the marginal conditions for which one of the bifurcates closes completely (full avulsion), and to compute the associated avulsion length $L_{AV}$. This is accomplished by setting $D_c = 0$ in Equation (25a) of our paper, and isolating the length as:

$$L_{AV} = \frac{D_0}{S_0} \frac{1 - \dfrac{\Delta\eta_{BRT}}{2}}{1 - S_b/S_0}. \tag{AC1}$$

The resulting expression highlights the key ingredients that determine the avulsion length, as all terms on the right-hand side of Equation (AC1) bring a precise physical meaning. Specifically, the first factor is the backwater length:

$$L_B = \frac{D_0}{S_0}, \tag{AC2}$$

which provides the length scale of $L_{AV}$. Furthermore, the numerator of the second factor accounts for the aggradation experienced by the non-dominant branch until it stops transporting sediment, while the denominator holds the contribution of the incision of the dominant branch. The latter term can be determined from Equation (25b), by setting $\Delta Q = 1$ and using Equation (24), in the form:

$$\frac{S_b}{S_0} = \frac{1}{2}\left(\frac{S_b}{S_0}\frac{D_b}{D_0}\right)^{3/2} = \frac{1}{2}\left(\frac{\theta_b}{\theta_0}\right)^{3/2}. \tag{AC3}$$

This allows us to re-write Equation (AC1) as

$$\frac{L_{AV}}{L_B} = \frac{1 - \dfrac{\Delta\eta_{BRT}}{2}}{1 - \dfrac{1}{2}\left(\dfrac{\theta_b}{\theta_0}\right)^{3/2}}, \tag{AC4}$$

where $\theta_b$ can be calculated from mass conservation (Equation (23) of our paper).

Results reported in Figure AC2b reveal that the order of magnitude of $L_{AV}$ is fixed by the backwater length. Moreover, as implied by Equation (AC4), the ratio $L_{AV}/L_B$ increases with increasing Shields stress and decreases with increasing $\beta_0$, due to the increase of the term $\Delta\eta_{BRT}$ (see Figure 10 of our paper).

2. The addition of the analytical model (25) is a noteworthy and helpful. But so that others can explore the model, I would suggest explicitly writing out the transport models that are used in the last component. The authors could also point towards what numerical method/tool they used to solve the system of nonlinear equations.

   Results reported in our paper have been obtained using the transport relationship of Meyer-Peter and Müller (1948) and that of Parker (1978), as mentioned in Section 2. The adopted transport model is reported in the respective caption of each figure: we will include any missing information in the revised version of the manuscript.
   To solve the nonlinear algebraic system, we simply use the default solver of the Python Scipy package. The solution seems to always converge, so we did not expect this choice to be critical.

3. Line 302: It is not clear to me what is meant by Beta_NT is calculated analytically. Is this arrived at by using the basic BRT analysis?

   The Reviewer is right. The $\beta_{NT}$ is calculated using the basic BRT analysis, as reported, for example, at the beginning of Section 3 (lines 231-232).

4. With reference to Fig 5. Why is there such an abrupt change (almost like a phase transition) at (or close to) Beta_NT. Is such a jump exhibited in the analytical model in (25)?

   The abrupt change pointed by the Reviewer emerges from the results of the numerical simulations while gradually increasing $\beta_0$ from values smaller than the no-transport threshold $\beta_{NT}$ to values larger than $\beta_{NT}$, as described in the first part of Section 3. Physically speaking, this abrupt change corresponds to a sharp transition in the system behaviour, as the non-dominant branch becomes unable to adapt its bedslope over time when its transport capacity vanishes. In this case, the BRT model is no longer applicable, as it assumes that both branches are morphodynamically active. This is why we developed a new analytical model (described in Section 4) to predict the long-term equilibrium state when $\beta_0 > \beta_{NT}$. Strictly

speaking, the analytical model defined by Equation (25) does not contain in itself any abrupt change. However, its applicability is obviously limited to cases where $\beta_0 > \beta_{NT}$.

**References**

Bertoldi, W., & Tubino, M. (2007). River bifurcations: Experimental observations on equilibrium configurations. *Water Resources Research*, *43*(10), 1–10. https://doi.org/10.1029/2007WR005907

Bolla Pittaluga, M., Repetto, R., & Tubino, M. (2003). Channel bifurcation in braided rivers: Equilibrium configurations and stability. *Water Resources Research*, *39*(3), 1–13. https://doi.org/10.1029/2001WR001112

Meyer-Peter, E., & Müller, R. (1948). Formulas for Bed-Load transport. *IAHSR 2nd Meeting, Stockholm, Appendix 2.*

Parker, G. (1978). Self-formed straight rivers with equilibrium banks and mobile bed. part 2. the gravel river. *Journal of Fluid Mechanics*, *89*, 127–146. https://doi.org/10.1017/S0022112078002505

Ragno, N., Redolfi, M., & Tubino, M. (2021). Coupled Morphodynamics of River Bifurcations and Confluences [Publisher: Blackwell Publishing Ltd]. *Water Resources Research*, *57*(1). https://doi.org/10.1029/2020WR028515

Ragno, N., Redolfi, M., & Tubino, M. (2022). Quasi-Universal Length Scale of River Anabranches. *Geophysical Research Letters*, *49*(16), e2022GL099928. https://doi.org/10.1029/2022GL099928

Salter, G., Voller, V. R., & Paola, C. (2019). How does the downstream boundary affect avulsion dynamics in a laboratory bifurcation? [Publisher: Copernicus GmbH]. *Earth Surface Dynamics*, *7*(4), 911–927. https://doi.org/10.5194/esurf-7-911-2019

Szewczyk, L., Grimaud, J.-L., & Cojan, I. (2020). Experimental evidence for bifurcation angles control on abandoned channel fill geometry [Publisher: Copernicus GmbH]. *Earth Surface Dynamics*, *8*(2), 275–288. https://doi.org/10.5194/esurf-8-275-2020

---

## Author Response (AR1)

**Authors' response to Reviewers**

We thank both Reviewers for the constructive comments and thoughtful suggestions. In the following we provide a detailed response to each point of the Reviewers' Comments, reporting the related changes to the manuscript where applicable. Please refer to the attached "diff.pdf" file for a complete list of all additions and minor changes to the manuscript.

The Reviewers' comments are in bold text, while our replies are in plain text. Changes made to the original version of the manuscript are in blue text.

**Reply on RC1**

**Overwiew**

**This work investigates the control of avulsions in bifurcated channel systems; a main channel splitting into 2 equal sub channels. Towards this end, the authors build a mathematical/numerical model, extending the well known analysis presented by Bolla Pittaluga et al. (2003) (BRT). The proposed model can track the evolution of a perturbation at the branching point of the sub-channels. In the first place, the model recovers the regimes identified in BRT. At low channel aspect ratios, following a perturbation, the system recovers balanced flows in each of the sub-channels. Beyond a critical aspect ratio Beta_c, however, the long term equilibrium of the flow in the system is unbalanced. This work show that as the aspect ratio is increased further, a second threshold Beta_TH is reached, here the sub channel, with the least flow, exhibits a partial-avulsion—where the channel still carries flow but ceases to transport sediment. Depending on the length of the sub channels, as the aspect ratio is increased even further, a point is reached where full avulsion occurs, i.e. the channel does not convey either water or sediment.**

**Comments**

1. **I think that the analysis in this paper is of sufficient interest to stand alone but also feel that it would be significantly enhanced, if the authors can point toward experimental or field evidence of the behaviors predicted by the math. For example, referring to Figure (9), it appears, by my calculations, that full avulsion would be reached in a system with channels of water depth of 2 m, width 40 m, and length 1 km+. How common are such conditions in field settings? (eg. Wax Lake in Louisiana)? Are records of permanently avulsed channels seen in such systems? Are records of permanently avulsed channels seen in field systems with shorter channel lengths and smaller aspect ratios?**

   We thank the Reviewer for the suggestions. Unfortunately, the available datasets are not sufficient to allow a detailed verification of model results.

   Data from physical models are scarce, and experiments were mainly focused on the analysis of the effect of specific "forcing" factors. For example, the laboratory experiments by Salter et al. (2019) were tailored to depositional environments and reproduced bifurcations with prograding branches, while Szewczyk et al. (2020) analysed the influence on water discharge partition of changing the bifurcation angle, for values of the aspect ratio $\beta_0$ much smaller than the no-transport threshold $\beta_{NT}$. To the authors' knowledge, the laboratory experiments by Bertoldi and Tubino (2007) are the only ones that analyse how initially balanced "free" bifurcations develop unbalanced configurations, possibly leading to partial or full avulsion. Indeed, a specific analysis of their results seems to support our findings, as shown in Figure AC1 where we report, for each experimental run, the computed values

[Figure]

Figure AC1: Results from the analysis of experimental data by Bertoldi and Tubino (2007). Consistently with our theory, experiments where full avulsions were observed (filled markers) fall in the region $\beta_0 > \beta_{NT}$ and $L > L_{AV}$.

of $\beta_0/\beta_{NT}$ and $L/L_{AV}$. Consistently with our predictions, they reveal that full avulsions (i.e. the complete closure of one of the branches) were observed in the three experimental runs in which $\beta_0$ was greater than (or closer to) $\beta_{NT}$ and the channel length $L$ was longer than the avulsion length $L_{AV}$.

To include this analysis in the main paper, we added the following paragraph to Sect. 5 (lines 456-464 of the manuscript):

"A direct comparison between our model and experimental data is difficult, since available data from physical models are scarce and often focused on the effect of external forcing factors. For example, the laboratory experiments by Salter et al. (2019) were tailored to depositional environments and reproduced bifurcations with prograding branches, while Szewczyk et al. (2020) analysed the influence on the water discharge partition of changing the bifurcation angle, for values of the aspect ratio $\beta_0$ much smaller than the no-transport threshold $\beta_{NT}$. To the authors' knowledge, the laboratory experiments by Bertoldi and Tubino (2007) are the only ones that analysed how initially balanced "free" bifurcations develop unbalanced equilibrium configurations, possibly leading to partial or full avulsion. Considering this dataset, it is worth noting that the three experimental runs that resulted in a full avulsion featured an aspect ratio close to or larger than the no-transport aspect ratio $\beta_{NT}$ and a length of the branches larger than the avulsion threshold $L_{AV}$, consistently with our model predictions."

Further support to our findings is also provided by the results of Ragno et al. (2022) who analysed nearly 200 bifurcation-confluence units of sand-bed and gravel-bed rivers. They found that field data show the existence of quasi-universal relations for the branches length when scaled with bankfull variables of the main upstream channel (the mean depth or width). The resulting average dimensionless length falls in the range $L^* = 200 - 300$,

[Figure]

Figure AC2: Dependence of (a) the dimensionless avulsion length $L_{AV}/D_0$ and (b) the scaled avulsion length $L_{AV}/L_B$ on the Shields stress $\theta_0$, for different values of the aspect ratio $\beta_0$, as obtained from Eq. (29). Vertical dashed lines indicate the value of Shields stress for which $\beta_0 = \beta_{NT}$, thus representing the boundary of the region of validity of the analytical model. The dotted line indicates the upper bound for $L_{AV}$, obtained by neglecting the $\beta_0$-dependent term $\Delta\eta_{BRT}$ ($C = 12$, transport formula by Meyer-Peter and Müller, 1948). This figure has been added to the main paper as the new Figure 10.

which can be interpreted as a preferential range of length values that allows river loops to keep both branches active. The results reported in Figure AC2a suggest that these values are typically lower than the predicted dimensionless avulsion length $L_{AV}/D_0$. We note, however, that the application of our model to this data-set is not straightforward, as the presence of the confluence may have an important influence on loop stability and equilibrium configuration (Ragno et al., 2021). Therefore, a direct comparison with our model's results would require incorporating the effect of the confluence, and considering the peculiar hydrodynamical and morphodynamical characteristics of individual field cases.

Regarding the estimate of the avulsion length proposed by the Reviewer based on Fig. 9 of our paper, we observe that the figure does not include conditions of full avulsion, which would correspond to the intersection of the curves with the $\Delta Q = 1$ line (incidentally, we also note that we define the aspect ratio $\beta_0$ as the half-width to depth ratio). To make this point clearer, we hereby provide the mathematical procedure, based on the analytical model described in Sect. 4 of our work, to compute the avulsion length $L_{AV}$ as a function of the flow conditions in the upstream channel, namely the Shields stress $\theta_0$ and the channel aspect ratio $\beta_0$, provided the latter is larger than the no-transport aspect ratio $\beta_{NT}$. The following paragraphs have also been added in the revised version of the manuscript at the end of Sect. 4.

" The analytical model makes it possible to determine the marginal conditions for which one of the bifurcates closes completely (full avulsion), and to compute the associated avulsion length $L_{AV}$. This is accomplished by setting $D_c = 0$ in Eq. (25a), and isolating the length

as:

$$L_{AV} = \frac{D_0}{S_0} \frac{1 - \dfrac{\Delta\eta_{BRT}}{2}}{1 - S_b/S_0}. \tag{26}$$

The resulting expression highlights the key ingredients that determine the avulsion length, as all terms on the right-hand side of Eq. (26) bring a precise physical meaning. Specifically, the first factor is the backwater length:

$$L_B = \frac{D_0}{S_0}, \tag{27}$$

which provides the length scale of $L_{AV}$. Furthermore, the numerator of the second factor accounts for the aggradation experienced by the non-dominant branch until it stops transporting sediment, while the denominator holds the contribution of the incision of the dominant branch. The latter term can be determined from Eq. (25b), by setting $\Delta Q = 1$ and using Eq. (24), in the form:

$$\frac{S_b}{S_0} = \frac{1}{2}\left(\frac{S_b}{S_0}\frac{D_b}{D_0}\right)^{3/2} = \frac{1}{2}\left(\frac{\theta_b}{\theta_0}\right)^{3/2}. \tag{28}$$

This allows us to re-write Eq. (26) as

$$\frac{L_{AV}}{L_B} = \frac{1 - \dfrac{\Delta\eta_{BRT}}{2}}{1 - \dfrac{1}{2}\left(\dfrac{\theta_b}{\theta_0}\right)^{3/2}}, \tag{29}$$

where $\theta_b$ can be calculated from mass conservation (23)."

Figure AC2 shows the dependence of the avulsion length $L_{AV}$ on the aspect ratio $\beta_0$ and on the Shields stress $\theta_0$ obtained from Eq. (29). This figure has also been added to the main paper (as the new Figure 10), along with the following descriptive paragraph (lines 409-414):

"Results reported in Fig. 10a reveal that, for typical parameter values of gravel bed rivers, the avulsion length is in the order of several hundred times the upstream flow depth. Furthermore, its dimensionless value exhibits a decreasing trend with both the Shields stress $\theta_0$ and the aspect ratio $\beta_0$. The former inverse dependence is mainly driven by the role of the channel slope $S_0$, while the latter is due to the increase of the term $\Delta\eta_{BRT}$ with $\beta_0$. Interestingly, when we filter out the main effect of $S_0$ by scaling the avulsion length $L_{AV}$ with the backwater length $L_B$, the dependence of the avulsion length on the Shields stress is reversed, as shown by Fig. 10b."

2. **The addition of the analytical model (25) is a noteworthy and helpful. But so that others can explore the model, I would suggest explicitly writing out the transport models that are used in the last component. The authors could also point towards what numerical method/tool they used to solve the system of nonlinear equations.**

Results reported in our paper have been obtained using the transport relationship of Meyer-Peter and Müller (1948) and that of Parker (1978), as mentioned in Sect. 2. We made sure to report the adopted transport model in all figure captions where necessary.
To solve the nonlinear algebraic system, we simply used the default solver of the Python Scipy package. The solution seemed to always converge, so we do not expect this choice to be critical.

3. **Line 302: It is not clear to me what is meant by Beta_NT is calculated analytically. Is this arrived at by using the basic BRT analysis?**

   The Reviewer is right. The $\beta_{NT}$ is calculated using the basic BRT analysis, as reported, for example, at the beginning of Sect. 3 (lines 232-234). For the sake of clarity, we modified line 302 as in the following:

   "[...] the threshold value $\beta_{NT}$ computed analytically by means of the BRT model (see dotted lines in the figure). "

4. **With reference to Fig 5. Why is there such an abrupt change (almost like a phase transition) at (or close to) Beta_NT. Is such a jump exhibited in the analytical model in (25)?**

   The abrupt change pointed by the Reviewer emerges from the results of the numerical simulations while gradually increasing $\beta_0$ from values smaller than the no-transport threshold $\beta_{NT}$ to values larger than $\beta_{NT}$, as described in the first part of Sect. 3. To make this point clearer, we added a panel to Figure 6 showing the variation over time of the discharge asymmetry $\Delta Q$ of numerical simulations in which $\beta_0$ is larger than $\beta_{NT}$. Contrary to simulations in which $\beta_0 < \beta_{NT}$, their equilibrium value of $\Delta Q$ is considerably larger than that foreseen by the BRT model. This difference in the long-term discharge asymmetry corresponds to the discontinuity represented in Fig. 5 and pointed out by the Reviewer.

   Physically speaking, this abrupt change corresponds to a sharp transition in the system behaviour, as the non-dominant branch becomes unable to adapt its bedslope over time when its transport capacity vanishes. In this case, the BRT model is no longer applicable, as it assumes that both branches are morphodynamically active. This is why we developed a new analytical model (described in Sect. 4) to predict the long-term equilibrium state when $\beta_0 > \beta_{NT}$. Strictly speaking, the analytical model defined by Eq. (25) does not contain in itself any abrupt change. However, its applicability is obviously limited to cases where $\beta_0 > \beta_{NT}$.

**Reply on RC2**

**The study effectively overcomes the limit of the original BRT (2003) model for river bifurcation to account for the situation where one of the two branches reaches vanishing transport capacity. The finding of more asymmetrical flow distribution in those configurations agrees with what is commonly found through numerical simulations (i.e. Kleinhans et al., 2008). The numerical scheme presented is robust, however, it would be interesting to have some comments on how it responds to different kinds of perturbations (i.e. shape, position or magnitude) and, eventually, the effect in terms of morphodynamic timescales. The analytical model to study those conditions, even though strongly idealized, does its job in describing the configuration where the non-dominant branch is not able to adjust its riverbed anymore. It would be nice to see in future developments the inclusion of finer sediments to apply those considerations in low-land bifurcations.**

We thank the Reviewer for the insightful comments on our work. We agree with the Reviewer that the transient behaviour of the bifurcation may depend on the magnitude and shape of the initial perturbation. We did not perform a systematic analysis of the effect of the initial perturbation, but we expect the long-term equilibrium of the bifurcation - which is the main topic explored by our work - not to depend on it. As a matter of fact, the long-term discharge asymmetry displayed by numerical simulations matches that foreseen by the analytical models. In our work we focus on the behaviour of "free" bifurcations and we do not model the generative

processes that lead to their formation. Therefore, we assumed the simplest, less disturbed initial configuration, as needed to isolate the basic mechanisms. In this sense, our analysis is only telling part of the story, in which the bifurcation is unforced and weakly perturbed.

It is worth recalling that several previous works examined the role of various forcing factors, such as the curvature of the upstream channel (Kleinhans et al., 2008), branches progradation (Salter et al., 2018), tides (Ragno et al., 2020) and slope advantage in the downstream branches (Redolfi et al., 2019). Furthermore, Bertoldi et al. (2009) highlighted how cyclic discharge variations or the complete abandonment of one branch can be the consequence of the migration of bars in the upstream channel.

We thank the Reviewer for the suggestion regarding the inclusion of finer sediment. We also believe that this would be an important future development. Therefore, we added the following paragraph at the end of Section 5.

"The present formulation is valid under conditions where sediment is dominantly transported as bedload. Although some promising approaches have been proposed (Iwantoro et al., 2021), a systematic understanding of the morphodynamics of channel bifurcations in suspension-dominated channels is still lacking, thus preventing a reliable extension of our model. However, we can expect that a similar scenario of transition to the avulsion condition also occurs for sand bed channels where most sediments are transported in suspension. In this case the higher values of the Shields stress are likely to be associated with an higher threshold $\beta_{NT}$, so that larger values of the aspect ratio may be needed to reach avulsion conditions."

**References**

Bertoldi, W., & Tubino, M. (2007). River bifurcations: Experimental observations on equilibrium configurations. *Water Resources Research*, *43*(10), 1–10. https://doi.org/10.1029/2007WR005907

Bertoldi, W., Zanoni, L., Miori, S., Repetto, R., & Tubino, M. (2009). Interaction between migrating bars and bifurcations in gravel bed rivers. *Water Resources Research*, *45*(6), 1–12. https://doi.org/10.1029/2008WR007086

Bolla Pittaluga, M., Repetto, R., & Tubino, M. (2003). Channel bifurcation in braided rivers: Equilibrium configurations and stability. *Water Resources Research*, *39*(3), 1–13. https://doi.org/10.1029/2001WR001112

Iwantoro, A. P., Vegt, M., & Kleinhans, M. G. (2021). Effects of sediment grain size and channel slope on the stability of river bifurcations. *Earth Surface Processes and Landforms*, *46*(10), 2004–2018. https://doi.org/10.1002/esp.5141

Kleinhans, M. G., Jagers, H. R., Mosselman, E., & Sloff, C. J. (2008). Bifurcation dynamics and avulsion duration in meandering rivers by one-dimensional and three-dimensional models. *Water Resources Research*, *44*(8). https://doi.org/10.1029/2007WR005912

Meyer-Peter, E., & Müller, R. (1948). Formulas for Bed-Load transport. *IAHSR 2nd Meeting, Stockholm, Appendix 2.*

Parker, G. (1978). Self-formed straight rivers with equilibrium banks and mobile bed. part 2. the gravel river. *Journal of Fluid Mechanics*, *89*, 127–146. https://doi.org/10.1017/S0022112078002505

Ragno, N., Redolfi, M., & Tubino, M. (2021). Coupled Morphodynamics of River Bifurcations and Confluences [Publisher: Blackwell Publishing Ltd]. *Water Resources Research*, *57*(1). https://doi.org/10.1029/2020WR028515

Ragno, N., Redolfi, M., & Tubino, M. (2022). Quasi-Universal Length Scale of River Anabranches. *Geophysical Research Letters*, *49*(16), e2022GL099928. https://doi.org/10.1029/2022GL099928

Ragno, N., Tambroni, N., & Bolla Pittaluga, M. (2020). Effect of Small Tidal Fluctuations on the Stability and Equilibrium Configurations of Bifurcations [Publisher: Blackwell

Publishing Ltd]. *Journal of Geophysical Research: Earth Surface*, *125*(8), 1–20. https://doi.org/10.1029/2020JF005584

Redolfi, M., Zolezzi, G., & Tubino, M. (2019). Free and forced morphodynamics of river bifurcations. *Earth Surface Processes and Landforms*, *44*(4), 973–987. https://doi.org/10.1002/esp.4561

Salter, G., Paola, C., & Voller, V. R. (2018). Control of Delta Avulsion by Downstream Sediment Sinks. *Journal of Geophysical Research: Earth Surface*, *123*(1), 142–166. https://doi.org/10.1002/2017JF004350

Salter, G., Voller, V. R., & Paola, C. (2019). How does the downstream boundary affect avulsion dynamics in a laboratory bifurcation? [Publisher: Copernicus GmbH]. *Earth Surface Dynamics*, *7*(4), 911–927. https://doi.org/10.5194/esurf-7-911-2019

Szewczyk, L., Grimaud, J.-L., & Cojan, I. (2020). Experimental evidence for bifurcation angles control on abandoned channel fill geometry [Publisher: Copernicus GmbH]. *Earth Surface Dynamics*, *8*(2), 275–288. https://doi.org/10.5194/esurf-8-275-2020